

# Effects of high-intensity sprint exercise on neuromuscular function in sprinters: the countermovement jump as a fatigue assessment tool

Takahiro Hasegawa[1,2], Kotaro Muratomi[3], Yuki Furuhashi[3], Jun Mizushima[4] and Hirohiko Maemura[5]

[1] Hillside Teachers' College, Bulawayo, Zimbabwe
[2] Japan Overseas Cooperation Volunteers, Japan International Cooperation Agency, Tokyo, Japan
[3] Graduate School of Comprehensive Human Sciences, University of Tsukuba, Ibaraki, Japan
[4] Faculty of Health and Sports Sciences, Toyo University, Tokyo, Japan
[5] Faculty of Health and Sport Sciences, University of Tsukuba, Ibaraki, Japan

## ABSTRACT

**Background.** High-intensity sprint exercises (HIS) are central to sprinter training and require careful monitoring of athlete muscle fatigue to improve performance and prevent injury. While the countermovement jump (CMJ) may be used to monitor neuromuscular fatigue (NMF), little is known about the specific effects from HIS. The purpose of this study is to investigate the effects of HIS on the CMJ to assess its utility for assessing NMF following HIS.

**Methods.** Ten male collegiate 400 m sprinters completed a 400 m sprint fatigue protocol and underwent five CMJ-testing sessions (baseline, 3 minutes, 10 minutes, 1 hour and 24 hours) over two days. Three CMJ trials, performed on a force plate, were completed each trial, with rating of perceived exertion (RPE) recorded as a subjective fatigue measure. Changes in RPE, CMJ variables, force-time and power-time curves at baseline and post fatigue were assessed.

**Results.** Significant changes were observed in most variables following the fatigue protocol. In particular, concentric mean power remained significantly lower after 24 hours compared to baseline. In addition, the force-time curves exhibited a significant reduction in all conditions following the fatigue protocol. This decline was most pronounced within 50–75% of the concentric phase relative to baseline measurements.

**Conclusion.** Results indicate that the CMJ may be a useful tool for monitoring fatigue in at least 400 m sprinters. These data also indicate that HIS may disproportionately reduce force output in during concentric movement. These insights may improve training prescriptions and injury prevention strategies for sprint athletes.

## INTRODUCTION

Sprint running events in athletics epitomize the pursuit of peak human performance, challenging athletes to cover set distances in the shortest possible time. Sprinter primarily perform high-intensity sprint exercises during their daily training and competitions,

Corresponding author
Takahiro Hasegawa,
hasegawan.tsukuba@gmail.com

typically ranging from 80 to 100% of their maximum sprint velocity (*Haugen et al., 2019*). Meticulously designed long-term training plans are then necessary to manage the above training and optimize sprint performance (*Haugen et al., 2019*). A critical aspect of these training programs is the assessment of internal load (*i.e.,* athletes' psychophysiological response to training), using both subjective and objective measurements (*Impellizzeri, Marcora & Coutts, 2019*). Such assessments are crucial for not only enhancing performance but also preventing training-related injuries or illness (*McGuigan, 2017*). However, there remains a lack of consensus on the most effective methods for evaluating training adaptations in sprinters, including subjective measures, heart rate monitoring, blood lactate levels and jumping exercises to assess the power capability (*Suzuki et al., 2006*; *Jimenez-Reyes et al., 2016*; *Cristina-Souza et al., 2019*; *Coyne et al., 2021*).

Neuromuscular fatigue is part of the broader concept of fatigue and refers to a reduction in maximum voluntary contractile force. It is result of deficits within the central nervous system, in the neural drive to the muscle, or within the muscle itself (*McGuigan, 2017*). The countermovement jump (CMJ) stands out as an objective assessment tool, widely recognized for its utility in monitoring neuromuscular function across diverse sports disciplines and environments (*Claudino et al., 2017*). The CMJ's appeal lies in its simplicity, the established correlation between CMJ performance and athletic performance in various sports, and its proven validity (*Gathercole et al., 2015c*; *McGuigan, 2017*). This has led to its widespread adoption for assessing neuromuscular function in diverse sports contexts (*Taylor et al., 2012*).

In general, muscle fatigue leads to a reduction in maximum voluntary contractile force that muscles can generate. Consequently, CMJ analysis has traditionally focused on overall metrics such as peak and mean values related to the jump's concentric phase, with particular attention to jump height and peak power (*Cormack et al., 2008a*; *Jimenez-Reyes et al., 2016*). However, this focus may not fully capture nuanced neuromuscular changes associated with muscle fatigue (*Gathercole et al., 2015a*; *Knicker et al., 2011*). *Taylor (2012)* found that variables related to the eccentric phase of the CMJ were the most indicative of muscle fatigue following continuous resistance training interventions. In addition, acute fatigue from continuous vertical jumps was shown to reduce lower limb flexion during the braking phase of subsequent CMJ (*Rodacki, Fowler & Bennett, 2001*; *McNeal, Sands & Stone, 2010*). This realization has prompted *Gathercole et al. (2015a)*; *Gathercole, Sporer & Stellingwerff (2015b)* to advocate for more comprehensive approach, incorporating variables from both the eccentric and concentric phases of the CMJ and considering movement strategies. Further expanding this perspective, recent studies (*Philpott et al., 2021*; *Hughes et al., 2022*; *Thomas, Jones & Dos'Santos, 2022*) have delved into analyzing CMJ variables using ground reaction force and examining the force-time and power-time curves, thereby offering a deeper understanding of muscle fatigue, variations in force exertion characteristics, and gender differences. *Hughes et al. (2022)* particularly emphasized the value of statistical parametric mapping (SPM) in analyzing the force-time curve obtained from the CMJ, shedding light on alterations in movement strategies induced by muscle fatigue. These methods may potentially be used to assess muscle fatigue in sprinters. However, research specifically targeting sprinters remains scarce. *Jimenez-Reyes et al. (2016)* explored muscle

fatigue in sprint training using the CMJ, focusing solely on jump height. Such a singular measure does not provide a comprehensive understanding of the muscle fatigue induced by high-intensity sprint exercise.

Therefore, this study aims to comprehensively explore the effects of high-intensity sprint exercise on neuromuscular function, utilizing the CMJ as an assessment tool for sprint athletes. By elucidating these effects, the findings may contribute to our understanding of optimal training loads and appropriate training interventions, ultimately aiming to minimize the risk of injuries among sprint athletes. We hypothesize that high-intensity sprint exercise significantly influences both the eccentric and concentric phases of the CMJ due to the high intensity of the exercise.

## METHODS

### Experimental designs

Our study employed a two-day experimental design, incorporating a single fatigue protocol and five CMJ-testing sessions to investigate the effects of high-intensity sprint exercise on neuromuscular function as assessed by the CMJ (Fig. 1).

Day 1 involved baseline CMJ measurement, followed by the high-intensity sprint exercise as the fatigue protocol (details of which are elaborated in a subsequent section). Neuromuscular function was then assessed using the CMJ at 3 min, 10 min, and 1 h after the fatigue protocol to capture immediate and short-term recovery responses. The Day 2 included a final CMJ assessment conducted 24 h after fatigue protocol to assess the recovery status.

### Participants and familiarization

Ten male university sprint athletes specializing in 400 m (age: 21.6 ± 1.5 years; height: 174.2 ± 2.9 cm; weight: 65.7 ± 5.0 kg; personal record: 48.87 ± 1.60 s) participated in the study. Throughout the testing period, participants refrained from engaging in any exercise activities other than this study's protocol. Comprehensive oral and written explanations of the study, including its purpose, procedures, and potential risks were provided. Informed consent was obtained in writing, and the study protocol was approved by the Ethics Committee of the Faculty of Health and Sports Sciences at the University of Tsukuba (IRB ID: tai 022-68).

Participants underwent a single CMJ practice session one week prior to the experiment to ensure familiarity with the proper technique. They received visual demonstrations and were instructed to focus on 'squatting quickly and jumping as high as possible'. Each participant completed 8–10 repetitions until CMJ technique was performed as consistently as possible. There was no restriction imposed on the depth of the squat.

### Fatigue protocol

The fatigue protocol for this study involved a 400 m sprint as the high-intensity sprint exercise, chosen to replicate sprint velocity and duration typically experienced by competitive sprinters. The 400 m was performed on the rubber surface of an outdoor track to closely simulate the conditions under which sprint-induced muscle fatigue occurs.

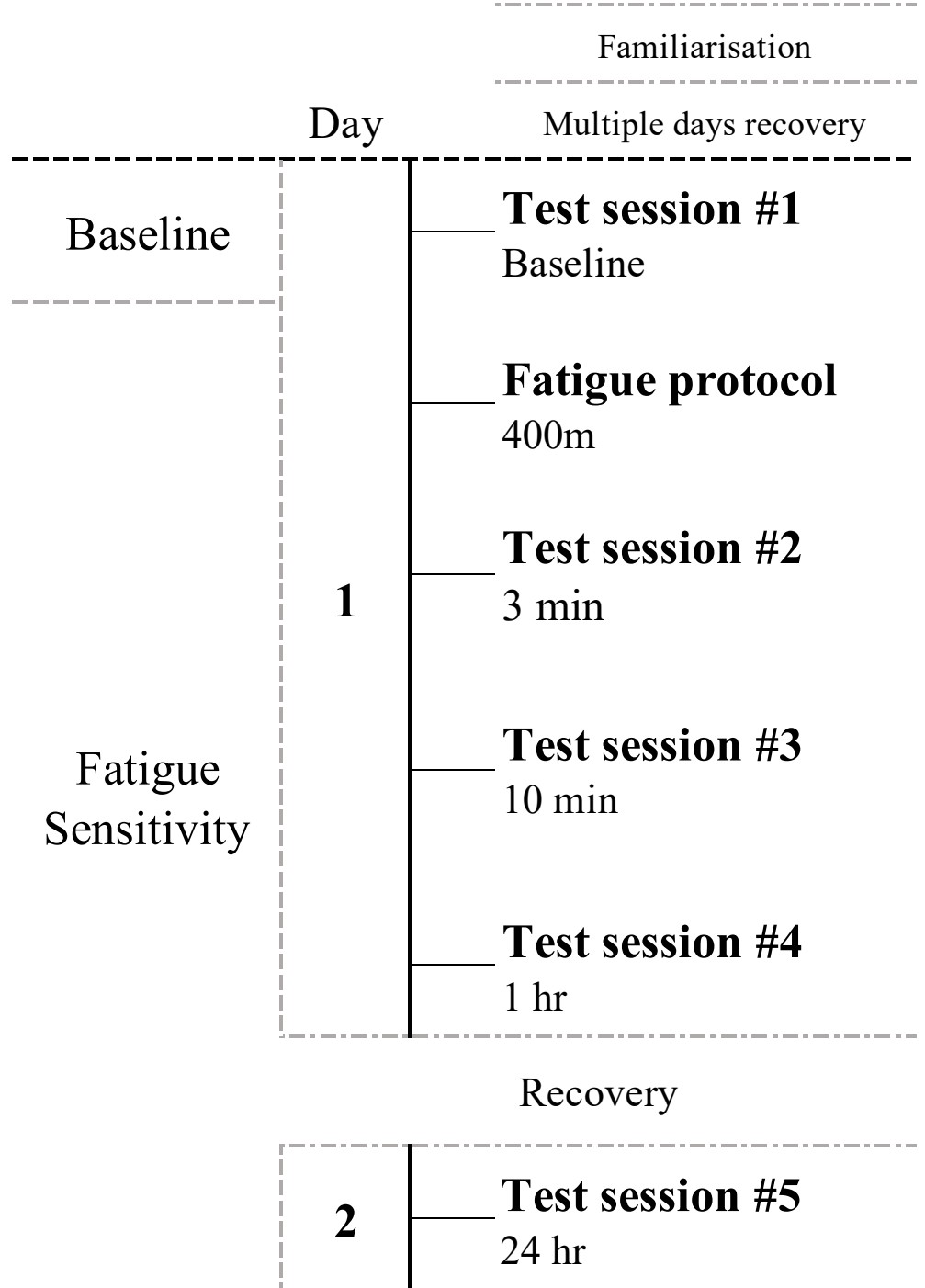

**Figure 1** Schematic representation of the study timeline including familiarization, fatigue protocol and fatigue-sensitivity sections.

Participants performed the 400 m wearing spike shoes and initiated their sprint from starting blocks at the sound of a pistol signal. The sprint was captured using a high-speed video camera (HC-WX2M; Panasonic, Tokyo, Japan) at a frame rate of 120 frames per second. The 400 m time was determined by timing from the appearance of smoke from the starter's pistol to the instant the participant's torso crossed the finish line.

### CMJ-Testing session

Participants performed three CMJ trials with 1.5 min of rest in between. Trials were performed on a force plate (1,000 Hz; Kistler, Winterthur, Switzerland) and sampled at 1,000 Hz using dedicated software (Ex-Jumper T, DKH, Tokyo, Japan) to obtain ground reaction force data. Calibration was performed for each trial to ensure accuracy by minimizing the deviation between actual values and measurement results. All trials were performed with their hands on hips to negate upper limb influence.

We also recorded a rating of perceived exertion (RPE) as the participant's subjective level of whole-body fatigue during each CMJ-testing session. The rating was based on the Borg scale (*Borg, 1982*), which has 15 levels ranging from 6 to 20. Participants visited the research facility at the same pre-determined time ($\pm$ 1.5 h) and participated in the CMJ-testing session on a total of five times.

### Baseline measurement

Participants, wearing training shoes, performed a 20-minute warm-up consisting of light jogging ($\sim$5 min), dynamic stretching, and a 50 m sprint. After the warm-up, participants performed five CMJ practices. Ten minutes later, the baseline CMJ-testing session began (Fig. 1).

### Fatigue sensitivity measurement

At 3 and 10 min after the fatigue protocol, three CMJ practices were performed before each testing session. At 1 h, a 10-minute warm-up consisting of dynamic stretching and three CMJ practices was performed. At 24 h, the same warm-up as at baseline was performed.

### Countermovement jump variables

Ground reaction force data obtained from the CMJ were categorized into distinct phases (*i.e.,* eccentric phase and concentric phase) according to the method of *Chavda et al. (2018)* (Fig. 2). CMJ variables were then calculated for each phase using Microsoft Excel (Microsoft, Redmond, WA, USA). These CMJ variables are described in Table 1. In this study, force, impulse, and power were normalized to the body mass of each participant. To ensure the validity and reliability of these variables, we averaged the values from three CMJ trials and represented each participant's performance per session (*Taylor et al., 2010*). The force-time and power-time curves the CMJ were normalized to 50% of the time in the eccentric phase (from the start point to the end of braking point) and 50% of the time in the concentric phase (from the end of braking point to the take-off).

### Statistical analysis

We conducted statistical analysis on RPE and CMJ variables utilizing IBM SPSS version 25 (SPSS Statistics, IBM, NY, USA). We employed a one-way analysis of variance (ANOVA)

 

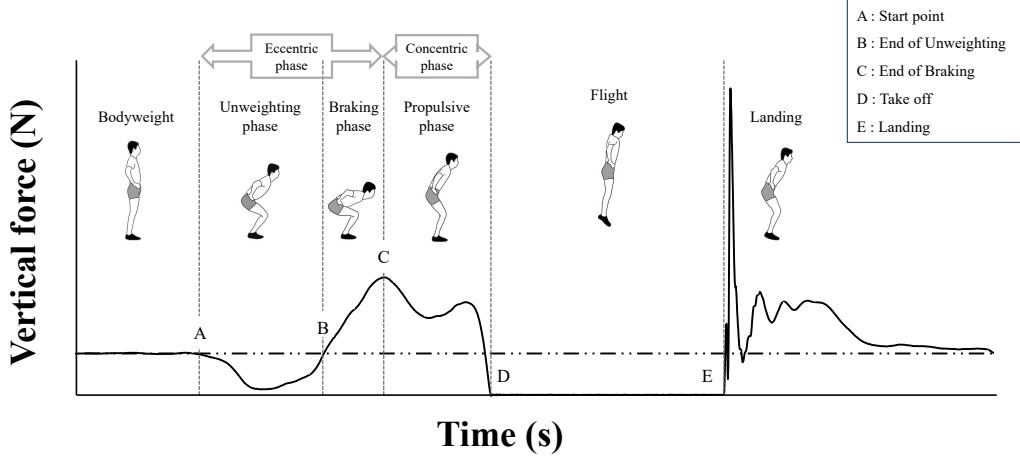

**Figure 2** **Force-time curve example of counter movement jump (CMJ) and definition of phase division.** This figure was created by the authors based on *Chavda et al. (2018)*.

**Table 1** **Description of jump height and CMJ variables.**

| Variable | Abbreviation | Description |
|---|---|---|
| Jump height (cm) | JH | The maximum jump height achieved, calculated using peak velocity |
| Peak force (N/kg) | PF | Greatest force achieved during the jump |
| Peak power (W/kg) | PP | Greatest power achieved during the jump |
| Eccentric impulse (Ns/kg) | EccI | Force exerted eccentrically multiplied by the time taken eccentrically |
| Concentric impulse (Ns/kg) | ConI | Force exerted concentrically multiplied by the time taken concentrically |
| Eccentric mean power (W/kg) | EccMP | Mean power generated during the eccentric phase of the jump |
| Concentric mean power (W/kg) | ConMP | Mean power generated during the concentric phase of the jump |
| Eccentric duration (s) | EccDur | Time of eccentric contraction during the jump |
| Concentric duration (s) | ConDur | Time of concentric contraction during the jump |
| Eccentric duration: Concentric duration (Time) | ED: CD | The ratio of eccentric duration to concentric duration |

to explore the main effects across the conditions: baseline, 3 minutes (3 min), 10 minutes (10 min), 1 hour (1 hr), and 24 hours (24 hr) after the fatigue protocol. Where the assumption of sphericity was violated as indicated by Mauchly's test, we applied the Greenhouse-Geisser correction. In instances where main effects reached statistical significance, we performed Bonferroni's post hoc tests to discern between-condition differences. Effect sizes were expressed using Partial Eta2 (squared) ($\eta^2$) values, with thresholds for small (0.04), moderate (0.25), and large (0.64) effects as recommended by *Ferguson (2009)*. The level of statistical significance was set at $P < 0.05$.

Additionally, we employed Statistical Parametric Mapping (SPM) with a paired $t$-test to compare the force-time and power-time curves of the CMJ with baseline and post-fatigue protocol conditions. SPM analysis was conducted using the spm1d code ($v.$ M0.1, http://www.spm1d.org) in MATLAB (Mathworks, Natick, MA, USA). A significance threshold for the SPM{t*} was set at $P < 0.05$.

# RESULTS

## Fatigue protocol

Participants completed the 400 m sprint as the fatigue protocol in $50.59 \pm 3.0$ s on average, which is 96.8% of their personal bests. The mean sprint velocity throughout the race was $7.93 \pm 0.45$ m/s.

## CMJ variables

Table 2 show the mean and SD of RPE and CMJ variables at baseline, 3 min, 10 min, 1 hr, and 24 hr. All variables were affected by the fatigue protocol and a significant main effect of condition was observed for these variables. Baseline RPE values were significantly lower than at 3 and 10 min (3 min: $p = 0.001$, 10 min: $p = 0.001$). Additionally, RPE at 3 min was significantly higher than at 10 min, 1 hr and 24 hr (10 min: $p = 0.02$, 1 hr: $p = 0.001$, 24 hr: $p = 0.001$). Further, RPE at 10 min was significantly higher than at 1 hr and 24 hr (1 hr: $p = 0.001$, 24 hr: $p = 0.001$). RPE at 24 hr was 11.5% lower than baseline.

Regarding jump height (JH), baseline values were significantly higher than at 3 and 10 min (3 min: $p = 0.003$, 10 min: $p = 0.013$). Moreover, JH at 3 min was significantly lower than at 10 min, 1 hr and 24 hr (10 min: $p = 0.004$, 1 hr: $p = 0.002$, 24 hr: $p = 0.002$). Similarly, JH at 10 min was significantly lower than at 1 hr and 24 hr (1 hr: $p = 0.016$, 24 hr: $p = 0.006$). JH at 24 hr was 5.6% lower than baseline.

For peak force (PF), baseline values were significantly higher than at 3 min and 24 hr (3 min: $p = 0.033$, 24 hr: $p = 0.033$), and PF at 3 min was significantly lower than at 10 min ($p = 0.022$). PF at 24 hr was 11.4% lower than baseline. As for peak power (PP), baseline values were significantly higher than at 3 min, 10 min and 24 hr (3 min: $p = 0.004$, 10 min: $p = 0.013$, 24 hr: $p = 0.011$), and PP at 3 min was significantly lower than at 10 min, 1 hr and 24 hr (10 min: $p = 0.006$, 1 hr: $p = 0.017$, 24 hr: $p = 0.014$). PP at 24 hr was 7.4% lower than baseline ($p = 0.011$).

Eccentric impulse (EccI) values at 3 min were significantly lower than at 10 min, 1 hr and 24 hr (10 min: $p = 0.029$, 1 hr: $p = 0.005$, 24 hr: $p = 0.007$). EccI at 24 hr was 0.1% higher than baseline. Eccentric mean power (EccMP) at 3 min was significantly lower than at 1 hr and 24 hr (1 hr: $p = 0.049$, 24 hr: $p = 0.006$). EccMP at 24 hr was 2.5% higher than baseline.

Regarding concentric impulse (ConI), baseline values were significantly higher than at 3 and 10 min (3 min: $p = 0.004$, 10 min: $p = 0.011$), with values at 3 min being lower than 1 hr and 24 hr (1 hr: $p = 0.014$, 24 hr: $p = 0.005$). Additionally, ConI at 10 min was significantly lower than at 24 hr ($p = 0.003$). ConI at 24 hr was 2.9% lower than baseline. For concentric mean power (ConMP), baseline values were significantly higher than at 3 min, 10 min and 24 hr (3 min: $p = 0.002$, 10 min: $p = 0.005$, 24 hr: $p = 0.009$), with values

**Table 2** Comparisons of RPE, JH and CMJ variables at baseline, 3 minutes (3 min), 10 minutes (10 min), 1 hour (1 hr) and 24 hours (24 hr) after fatigue protocol.

| | Baseline Mean ± SD | | 3 min Mean ± SD | | 10 min Mean ± SD | | 1 hr Mean ± SD | 24 hr Mean ± SD | Main Effect F (4,45) | Effect size (partial $\eta^2$) |
|---|---|---|---|---|---|---|---|---|---|---|
| RPE (a.u.) | 10.40 ± 3.03 | [a,b] | 18.80 ± 1.14 | [e,f,g] | 16.60 ± 1.51 | [h,i] | 11.00 ± 2.40 | 9.20 ± 3.26 | 50.03* | 0.85 |
| JH (cm) | 49.03 ± 6.00 | [a,b] | 34.10 ± 6.54 | [e,f,g] | 41.09 ± 4.55 | [h,i] | 45.96 ± 4.02 | 46.27 ± 3.61 | 26.84* | 0.75 |
| PF (N/kg) | 17.28 ± 2.26 | [a,d] | 14.43 ± 2.12 | [e] | 15.95 ± 2.20 | | 15.61 ± 2.34 | 15.31 ± 1.76 | 7.12* | 0.44 |
| PP (W/kg) | 65.19 ± 7.01 | [a,b,d] | 50.19 ± 5.49 | [e,f,g] | 57.03 ± 3.99 | | 61.18 ± 6.20 | 60.34 ± 4.86 | 20.41* | 0.69 |
| EccI (Ns/kg) | 1.35 ± 0.28 | | 1.16 ± 0.25 | [e,f,g] | 1.28 ± 0.31 | | 1.31 ± 0.23 | 1.35 ± 0.19 | 7.57* | 0.46 |
| ConI (Ns/kg) | 3.08 ± 0.24 | [a,b] | 2.51 ± 0.33 | [f,g] | 2.78 ± 0.20 | [i] | 2.94 ± 0.16 | 2.99 ± 0.17 | 21.60* | 0.71 |
| EccMP (W/kg) | 6.53 ± 1.43 | | 5.75 ± 1.46 | [f,g] | 6.40 ± 1.63 | | 6.38 ± 1.20 | 6.70 ± 1.16 | 7.03* | 0.44 |
| ConMP (W/kg) | 35.89 ± 4.04 | [a,b,d] | 24.59 ± 4.25 | [e,f,g] | 30.00 ± 2.29 | | 32.61 ± 3.07 | 32.42 ± 2.62 | 26.70* | 0.75 |
| EccDur (s) | 0.35 ± 0.05 | [b] | 0.33 ± 0.04 | | 0.32 ± 0.04 | | 0.34 ± 0.05 | 0.34 ± 0.04 | 4.27* | 0.32 |
| ConDur (s) | 0.25 ± 0.04 | [a,b,c,d] | 0.28 ± 0.04 | | 0.26 ± 0.04 | | 0.27 ± 0.03 | 0.27 ± 0.03 | 8.74* | 0.49 |
| ED: CD (Time) | 0.73 ± 0.14 | [a,b,d] | 0.86 ± 0.16 | | 0.83 ± 0.13 | | 0.81 ± 0.16 | 0.83 ± 0.15 | 11.66* | 0.56 |

**Notes.**

Significant difference between conditions ($p < 0.05$).

[a] baseline *vs* 3 min.
[b] baseline *vs* 10 min.
[c] baseline *vs* 1 hr.
[d] baseline *vs* 24 hr.
[e] 3 min *vs* 10 min.
[f] 3 min *vs* 1 hr.
[g] 3 min *vs* 24 hr.
[h] 10 min *vs* 1 hr.
[i] 10 min *vs* 24 hr.
[j] 1 hr *vs* 24 hr.
*Significant main effect ($p = 0.001$).

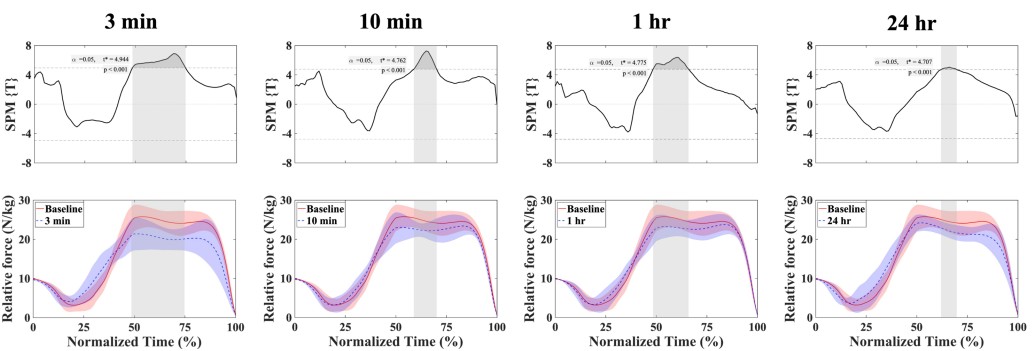

**Figure 3** **Mean ± SD relative force and SPMt\* output statistics in the CMJ for baseline and fatigue conditions (3 min, 10 min, 1 hr, and 24 hr).** Where the SPMt\*curve exceeds the critical threshold (dotted line), this area is shaded and a statistically significant differences exist at those nodes with $p$ values provided for each supra-threshold cluster. baseline = red; fatigue conditions = blue.

at 3 min being significantly lower than at 10 min, 1 hr and 24 hr (10 min: $p = 0.002$, 1 hr: $p = 0.006$, 24 hr: $p = 0.009$). ConMP at 24 hr was 9.7% lower than baseline ($p = 0.009$).

Eccentric duration (EccDur) at baseline was significantly higher than 10 min ($p = 0.022$). EccDur at 24 hr was 3.4% lower than baseline. Regarding concentric duration (ConDur), the baseline being significantly lower than at all subsequent conditions (3 min: $p = 0.005$, 10 min: $p = 0.018$, 1 hr: $p = 0.034$, 24 hr: $p = 0.015$). ConDur at 24 hr was 10.1% higher than baseline ($p = 0.015$). In terms of the ratio of eccentric duration:concentric duration (EC:CD), baseline values were significantly lower than at 3 min, 10 min and 24 hr (3 min: $p = 0.001$, 10 min: $p = 0.001$, 24 hr: $p = 0.019$). ED:CD at 24 hr was 14.0% higher than baseline ($p = 0.019$).

### Force-time curve and power-time curve in CMJ

Figs. 3 and 4 display the results of the force-time curve and the power-time curve, respectively, comparing baseline measurements to those taken at 3 min, 10 min, 1 hr and 24 hr after the fatigue protocol. For the force-time curve, there were significant differences observed between the baseline and each subsequent condition: between baseline and 3 min (48.6–74.8%), 10 min (58.9–70.2%), 1 hr (48.4–66.0%), and 24 hr (62.0–69.8%).

Concerning the power-time curve, significant variations were observed between the baseline and other conditions with the exception of the 1 hr. Specifically, there were notable differences between baseline and 3 min (56.1–90.0%), 10 min (65.6–87.6%), and 24 hr (65.9–88.2% and 98.3–99.4%).

## DISCUSSION

The purpose of this study was to comprehensively explore the effects of high-intensity sprint exercise on neuromuscular function, utilizing the CMJ as an assessment tool for sprint athletes. Our findings revealed that the 400 m sprint alters the ED:CD in the CMJ and significantly reduces force output in during concentric movement. On the other hand, the eccentric phase variables, such as EccI and EccMP, returned to their baseline after 10 min.

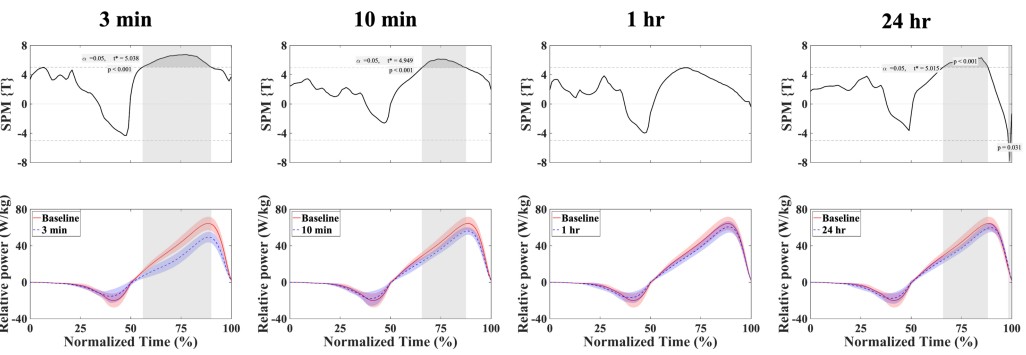

**Figure 4   Mean ± SD relative power and SPMt* output statistics in the CMJ for baseline and fatigue conditions (3 min, 10 min, 1 hr, and 24 hr).** Where the SPMt*curve exceeds the critical threshold (dotted line), this area is shaded and a statistically significant differences exist at those nodes with *p* values provided for each supra-threshold cluster. baseline = red; fatigue conditions = blue.

These findings confirm our hypothesis that high-intensity sprint exercise significantly impacts both the eccentric and concentric phases of the CMJ, highlighting its utility in sprinters' muscle fatigue monitoring.

## The importance of objective indicators

The 400 m sprint requires athletes to maintain high velocity throughout the race. This event is characterized as a prolonged sprint, largely due to the significant fatigue resulting from the glycolytic effort (*Zouhal et al., 2010*) and is considered to be one of the most demanding events in athletics (*Hanon & Gajer, 2009*). *Hirvonen et al. (1992)* compared the development of fatigue during the 400 m sprint with the 100 m, 200 m and 300 m sprints. They reported that a decrease in velocity occurred after 200 m and that at the end of 400 m, creatine phosphate stores were depleted, and lactate concentrations attained an individual maximum. Given these observations, the 400 m sprint, as a single sprint exercise, was proposed as a valid way for inducing muscle fatigue in sprinters.

In this study, some CMJ variables such as PP, ConMP, ConDur and CD: ED were significantly changed after 24 hr compared to baseline. This aligns with previous studies indicating that CMJ variables can vary over a period of several days after fatiguing exercise (*McLellan, Lovell & Gass, 2011*; *Cormack, Newton & McGuigan, 2008b*). Our findings support those observations, suggesting that a single 400 m sprint can lead to detectable muscle fatigue persisting over several days.

In assessing fatigue, we incorporated both subjective assessment of physical fatigue, using the RPE, and objective measures from each CMJ-testing session. Notably, our results showed deviations between the subjective RPE and objective CMJ variables such as PP and ConMP, especially evident at 24 h (see Table 2). These discrepancy suggests that while participants might subjectively feel recovered, the CMJ can detect residual neuromuscular fatigue, underlining the importance of objective measures in fatigue assessment.

Previous studies have highlighted the usefulness of subjective measures in assessing athlete fatigue (*Costa et al., 2022*; *Selmi et al., 2022*). However, our results underscore the need for a comprehensive approach that integrates both subjective and objective

evaluations. *Lourenço et al. (2023)* investigated this interplay between objective (*i.e.,* jumping exercise) and subjective fatigue assessments in football players, finding only a weak correlation between them. This suggests that subjective assessments might reflect more psychological and sociological aspects, emphasizing the need for multifaceted approach to assess muscle fatigue in athletes.

Therefore, we advocate for a combined use of subjective assessments and objective measures, such as the CMJ, to provide a more nuanced understanding of muscle fatigue, particularly in sprinters. Such an approach will enhance the precision of fatigue assessments and contribute to more effective training and recovery strategies for athletes.

### CMJ variables

The results of this study revealed significant changes in most CMJ variables following the fatigue protocol, as detailed in Table 2. Notably, several variables exhibited distinct recovery patterns. In particular, variables associated with the concentric phase, such as PP and ConMP, demonstrated a sustained decrease even after 24 h compared to baseline, suggesting prolonged effects of fatigue. In contrast, variables associated with the eccentric phase, including EccI and EccMP, showed a more rapid return to baseline levels.

This differentiation in recovery between the concentric and eccentric phases aligns with the findings of *Viitasalo et al. (1993)*, which noted that more pronounced effects of muscle fatigue in the concentric phase during jumping exercises. This is further supported by previous studies on low-velocity repetitive running such as the Yo-Yo test and exercises performed during rugby matches, which indicated a diminished ability for force exertion in the concentric phase following such exercises (*McLellan, Lovell & Gass, 2011*; *Gathercole et al., 2015a*). The findings of this study corroborate these previous studies, suggesting that high-intensity sprint exercises specifically challenge and potentially diminish the force-generating capacity of muscles during the concentric phase of movement.

On the other hand, EccI and EccMP returned to their baseline levels after 10 min. Moreover, after the fatigue protocol, EccDur decreased, while ConDur increased, leading to a higher ED:CD. This shift implies that participants performed the CMJ in a relatively concentric manner compared to baseline. *Rodacki, Fowler & Bennett (2001)* observed a 20% reduction in eccentric displacement following acute fatigue induced by repetitive CMJs, which was associated with less knee flexion, while hip and ankle angular displacement remained unchanged. Although this study did not directly measure lower limb joint displacements in CMJ, it is suggested that EccDur may shorten as a result of reduced displacement in the eccentric phase due to the effects of muscle fatigue.

Reducing the amount of knee flexion increases joint stiffness at the end of the eccentric phase of the jump (*Rodacki, Fowler & Bennett, 2001*). Such increased leg stiffness has been interpreted as a subconscious strategy employed to maintain or maximize stretch-shortening-cycle (SSC) performance under muscle fatigue. Furthermore, minimizing the eccentric phase during the CMJ under fatigue conditions could be related to the strategy of avoiding muscle damage caused by eccentric muscle contraction (*Rodacki, Fowler & Bennett, 2001*). It is suggested that the muscle fatigue induced by the 400 m sprint may have

altered the athlete movement strategy and contributed to an earlier recovery by maintaining the ability to exert force in the eccentric phase while avoiding greater muscle damage.

Another possible factor influencing the early recovery of this eccentric phase variables is the participants characteristics. Eccentric contraction generates greater forces for a given angular velocity than other contraction types (*Hortobágyi & Katch, 1990*), resulting in greater muscle damage. *Margaritelis et al. (2021)* examined the dynamics of creatine kinase, finding that muscle damage is triggered by muscle unaccustomedness in response to high-intensity eccentric exercise. Therefore, non-sprinters might experience more severe eccentric muscle damage following 400 m sprint. On the other hand, the participants in this study, accustomed to frequent sprint training, likely have a higher tolerance for eccentric loads on the muscles during sprinting. This adaptation may have contributed to less muscle damage and facilitated earlier recovery. Nonetheless, without the measurement of biomarkers such as creatine kinase to detect muscle damage, these conclusion remain some-what unclear.

## Force-time curve and power-time curve in CMJ

The force-time curves were significantly lower around 50–75% in all compared to baseline (3 min: 48.6–74.8%, 10 min: 58.9–70.2%, 1 hr: 48.4–66.0%, and 24 hr: 62.0–69.8%), and these effects were also observed at 1 and 24 hr (as shown in Fig. 3). Given that the 50% point is the end of braking point, which is the transition from the eccentric phase to the concentric phase (Fig. 2), it is suggested that high-intensity sprint exercise may affect the ability to exert force during the early part of concentric phase, especially in a more flexed position.

Similarly, the power-time curves showed a significant decrease compared to baseline from around 60% to around the peak power point (as shown in Fig. 4, 3 min: 56.1–90.0%, 10 min: 65.6–87.6%, and 24 hr: 65.9–88.2% and 98.3–99.4%). The phases in which significant differences were observed differed from those in the force-time curves. It is suggested that the reduced ability to exert force around the braking point may lead to the reduced ability to exert power up to around the peak power point in the concentric phase.

Previous studies have shown the effects of various exercise-induced muscle fatigue on the CMJ, particularly in the concentric phase and the SSC movement (*Rodacki, Fowler & Bennett, 2001*; *Nicol, Avela & Komi, 2006*; *Gathercole et al., 2015a*). Moreover, training in a fatigued state may increase the risk of injury (*Schwiete et al., 2023*). For example, to minimize the risk of injury in squats, it is necessary to use proper movement mechanics, such as keeping the lumbar spine neutral and avoiding excessive trunk flexion during the ascent (*Myer et al., 2014*). However, decreased force exertion in the flexed position under muscle fatigue may result in a loss of lumbar spine control. Furthermore, in an attempt to compensate for force, there may be a loss of proper movement mechanics by exerting force in an overstressed posture.

Consequently, this study suggests that post high-intensity sprint exercises, training activities demanding greater force exertion in a flexed position, such as weight training involving deeply flexed hip joints, might be reconsidered. Sprinters and their coaches should be aware of the implications of the posture, squat depth, and load in training sessions

following high-intensity sprint training, particularly when planning weight training or similar activities the following day.

## Usefulness of the CMJ test for sprinters

SSC muscle function during running is characterized by pre-activation to resist ground impact, followed by braking (the eccentric phase) and subsequent push-off (the concentric phase) (*Komi, 2000*). These actions result in a complex loading of the neuromuscular system (*Nicol, Avela & Komi, 2006*), involving metabolic, mechanical and neural components (*Komi, 2000*). It is therefore suggested that the 400 m sprint might have similarly complex effects on neuromuscular function.

Neuromuscular fatigue is broadly classified into peripheral and central components, based on the origin of fatigue mechanisms, either distally or proximally to the neuromuscular junction, respectively (*Gandevia, 2001*), with evidence suggesting their interdependence (*Jubeau et al., 2014*). Peripheral fatigue is defined as a reduction in the muscle force-generating capacity of skeletal muscles due to changes distal to the neuromuscular junction (*Ross et al., 2007*). Peripheral factors responsible for this reduced force capacity relate to metabolic changes such as altered intracellular milieu and depletion of energy substrates. Central fatigue involves an voluntary reduction in motor drive by the central nervous system, aiming to prevent catastrophic changes in homeostasis (*Ament & Verkerke, 2009*; *Gandevia, 2001*). Although reduced muscle function is thought to indicate fatigue, neuromuscular fatigue may also be manifested by qualitative changes in motor control (*Ament & Verkerke, 2009*). It is posited that neural changes might mitigate the effects of fatigue on muscle function by altering intra- and inter-limb strategies, such as through synergistic muscle activation, and load distribution among motor units (*Knicker et al., 2011*).

In this study, the decrease in variables like PP and ConMP, alongside changes in the ED:CD, indicate that the 400 m sprint could induce neuromuscular fatigue, including peripheral and central components. The CMJ is highlighted as a valuable tool for assessing not just force output but also changes in movement strategies, suggesting its efficacy as a comprehensive test for detecting neuromuscular fatigue in sprinter.

## LIMITATION

A limitation of this study is the somewhat narrow scope of the fatigue protocol employed, which may not fully encapsulate the variety of training modes typically utilized by sprinters. In this study, 400 m sprint was used as the fatigue protocol, but sprinters in actual training situations often engage in diverse trainings, including sprints at an individual's maximum velocity and resistance training using the slope of a hill (*Haugen et al., 2019*). Therefore, the specific muscle fatigue experienced in such varied training regimens may differ from what was induced and analyzed in our study. Thus, future research would need to understand the nuances of muscle fatigue induced by different modes of sprint training using the CMJ.

## CONCLUSION

Our results suggest that variables associated with the concentric phase, such as PP and ConMP, may be the best indicators of fatigue in trained sprinters. We also suggest that focusing on changes in movement strategy (*i.e.,* ED:CD) may enable a more comprehensive assessment of muscle fatigue. Furthermore, our use of SPM analysis provided deeper insights, suggesting that this reduced force capacity is especially notable in the early part of the concentric phase. Based on our findings, sprinters and their coaches can utilize the CMJ in the training field to assess muscle fatigue. Assessing the adaptation of the body to training by performing the CMJ once or twice a week may allow the prescription of an appropriate training load (*e.g.,* sprinting velocity, running distance). In addition, avoiding weight training with the lower limb joints in a more flexed position after high-intensity sprint exercise may contribute to reducing the risk of injury. Thus, CMJ may provide answers not only to the setting of a more optimal training load but also to the choice of training means for sprinter.

### Funding
The authors received no funding for this work.

### Competing Interests
Takahiro Hasegawa is employed by the Japan International Cooperation Agency.

### Author Contributions
- Takahiro Hasegawa conceived and designed the experiments, performed the experiments, analyzed the data, prepared figures and/or tables, authored or reviewed drafts of the article, and approved the final draft.
- Kotaro Muratomi conceived and designed the experiments, performed the experiments, analyzed the data, prepared figures and/or tables, authored or reviewed drafts of the article, and approved the final draft.
- Yuki Furuhashi performed the experiments, analyzed the data, authored or reviewed drafts of the article, and approved the final draft.
- Jun Mizushima analyzed the data, prepared figures and/or tables, authored or reviewed drafts of the article, and approved the final draft.
- Hirohiko Maemura conceived and designed the experiments, performed the experiments, analyzed the data, authored or reviewed drafts of the article, and approved the final draft.

### Human Ethics
The following information was supplied relating to ethical approvals (*i.e.,* approving body and any reference numbers):

The study protocol was approved by the Ethics Committee of the Faculty of Health and Sports Sciences at the University of Tsukuba (IRB ID: tai 022-68).

## Data Availability

The raw CMJ data is available in the Supplementary Files (baseline, 3min, 10min, 1hr, 24hr).

## Supplemental Information

Supplemental information for this article can be found online at http://dx.doi.org/10.7717/peerj.17443#supplemental-information.

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
