# Peer review of "Effects of high-intensity sprint exercise on neuromuscular function in sprinters: the countermovement jump as a fatigue assessment tool"

_PeerJ, doi:10.7717/peerj.17443_

## Round 0.1 · original submission · Major Revisions

Please attend to the reviewer´s comments. I think the manuscript is good but needs some improvements.

Regards.

Dr. Manuel Jimenez

·

Basic reporting

I attached generalized and specific comments. In short, the paper needs to be tightened and better focussed. There are significant questions in the attached review.

Experimental design

See attached comments.

Check this recent paper:

Article
Assessment of Countermovement Jump: What Should We Report?

Zdravko Anicic 1, Danica Janicijevic 1, Olivera M. Knezevic 1 , Amador Garcia-Ramos 2,3 , Milos R. Petrovic 1 ,
Dimitrije Cabarkapa 4
and Dragan M. Mirkov 1,*

Validity of the findings

See attached comments

Additional comments

See attached comments

·

Basic reporting

Overall, I found the paper to be well written and cited. The authors provide sufficient data from the study. My main concern and recommendations are 1) I did not see any hypotheses; you need to clearly articulate what your expected outcomes were. 2) Tie those hypotheses in with the opening of the discussion, what did you "hit" or "miss"? 3) I am not really seeing a clearly articulation of the relevance of Fig 5 and 6. 4) I'd like to see discussion of practical applications of how to utilize these findings in training or research.

Experimental design

I believe the design is explanatory and replaceable based on the information provided.

Validity of the findings

The results as outlined are sufficient. Based on my prior comments, I think the discussion could be enhanced with some more detail and some detail on how these new findings can be applied in practice or in further research of the topic.

Reviewer 3 ·

Basic reporting

This study is intriguing as it seeks to refine the evaluation of neuromuscular fatigue resulting from sprint exercise by employing the countermovement jump test. The findings not only offer valuable insights but also propose a method that could prove instrumental in monitoring athlete fatigue in practical settings.

This paper is acceptable for publication pending minor revisions. Namely the introduction could be improved by providing more literature references about muscular and neuromuscular fatigue generated by a sprint running to provide a better definition and clarify the aim of this study.

Experimental design

To ensure clarity, it's important to specify the precise objective of the study. Here are two potential objectives, each focusing on a distinct aspect:
- "Enhance understanding of the neuromuscular fatigue induced by a 400m sprint."
- "Improve the efficacy of the countermovement jump test as a measure of neuromuscular fatigue induced by sprint exercise."

By clearly defining the objective, the purpose and focus of the study become more apparent, facilitating better comprehension for readers.

Then, I think the tittle could be changed in a way to better understand the purpose.
For example :"Enhancing Neuromuscular Fatigue Assessment Following Sprint Exercise Through Improved Countermovement Jump Test".

Also, what distinguishes a 'high-intensity sprint' from a regular sprint? Is a sprint not inherently defined by maximal intensity?

Nonetheless, the method and protocol are well designed.

Validity of the findings

A comprehensive dataset has been compiled, featuring variables of significance to the field. The main variables, will be presented graphically, while others will be summarized in tabular format for clarity, such as RPE.

In the results section, lines 194 to 243, formulations like "multiple comparisons revealed" will be omitted before each result, as statistical analyses have been detailed previously, ensuring conciseness and clarity.

Regarding the discussion (lines 342-347), the link between "depth squats and posture", resistance training and injuries could be elaborated to avoid ambiguity and ensure coherence with the study's objectives and findings.

Also, to enhance the discussion section with a more mechanistic approach, consider delving deeper into the underlying physiological mechanisms that contribute to neuromuscular fatigue induced by sprint exercise and how the countermovement jump test serves as a proxy for assessing these mechanisms. This could involve discussing factors such as muscle activation patterns, energy substrate utilization, neural fatigue, or biomechanical changes during sprinting and subsequent jump performance.

---

## Round 0.2 · Minor Revisions

Dear Co-Authors,

Please read the reviews and consider modifying some of those comments.

Sincerely,

Dr. Manuel Jimenez

·

Basic reporting

The authors are commended for doing a thorough revision that addressed most major issues. There are two remaining areas of concern. First the abstract is too long. There is too much introduction in the opening and too much conclusion at its end, which best belongs in the conclusion of the discussion. This is an e-journal, so page numbers isn’t an issue, but typical abstracts are 200 to 250 words. This one is still 360 words. The second issue is the inclusion of reliability data. I understand that the authors want to make sure effort was similar across the jumps, but is that a practical component of using CMJ to assess fatigue post sprint work out?

In a practical sense a coach will do baseline measures and assume athlete does his best effort post workout as the coaches assess fatigue. I could be mistaken, but I don’t think any of the three reviewers of this manuscript would have asked or thought “did the athletes try hard on the jumps”. And as stated, it isn’t practical to have coaches make reliability measures. In short, the reliability aspect of the CMJ is a neat thought but not the aim the paper. Perhaps it is part of your next work? Is each component of the CMJ reliable. I wouldn’t introduce fatigue to assess that, though.

Intro:

L52: Sprint running epitomizes
Paragraph 1 as a whole seems more geared towards adaptation (L60). I think you are missing connections. Enhance performance and reduce injuries….to that end what are the variables that may impact these goals. List these rather than use L 60 and para one segues better to para 2.

L 105. May rather than can.

Methods:

L 140. Maybe competitive sprinters rather than athletes. I agree with reviewer 3. Sprints of any duration are maximal efforts. It is redundant to call them high intensity sprint exercise.

Results:

Reliability data are noise. Maybe pilot work for project 2.

Results are long. If this were MSSE, the editor would say no. I like words better than figures. Tables better than figures. Your words should add rather than repeat. But this is an e journal. So, there are no page charges.

Discussion:

L322: You didn’t measure any recovery processes. You observed patterns.

L349: The CMJ doesn’t have a strategy. The athlete may, however.

L453: Sprinters and their coaches.

Experimental design

Good

Validity of the findings

Good

Additional comments

See above

·

Basic reporting

I have provided most of my comments below. There are no obvious scientific flaws other than it appears the authors did not have any specific hypotheses at the inception. The structure of the paper is often disjointed and lacks logical flow. The major issue is excess. There's too much data presented to be easily reviewed, in part bc there's too much narrative. There's redundancy in data presentation in text, table and figures.

They've got to get very detailed on what the purposes and hypotheses were and clearly show how each was tested with the outcome, then focus on explaining that.

Experimental design

The overall design is adequate, but the excess data is not presented well.

Validity of the findings

The results appear valid and interesting.

Additional comments

I have attempted to revise the abstract to demonstrate how the authors can streamline the paper.

Background. High-intensity sprint exercises (HIS) are central to sprintertraining and require careful monitoring of athlete muscle fatigue to improveperformance and prevent injury. While the countermovement jump (CMJ) may be usedto monitor neuromuscular fatigue (NMF), little is known about the specific effectsfrom HIS. The purpose of this study is to investigate the effects of HIS on theCMJ to assess its utility for assessing NMF following HIS.Methods. Ten male collegiate400 m sprinters completed a 400 msprint fatigue protocol and underwent five CMJ-testing sessions (baseline, 3minutes, 10 minutes, 1 hour and 24 hours) over two days. Three CMJ trials, performedon a force plate, were completed each trial, with rating of perceived exertion(RPE) recorded as a subjective fatigue measure. Changes in RPE, jump height,CMJ variables, force-time and power-time curves at baseline and post fatigue wereassessed. Results. Significantchanges were observed in most variables following the fatigue protocol. Both RPEand jump height decreased significantly after the fatigue protocol, returning nearbaseline levels at 24 hours. However, while eccentric impulse and eccentricmean power also approached baseline by 24 hours, concentric mean power remainedsignificantly reduced. Recovery patterns differed among CMJ variables, notablyforce-time curves, which exhibited a significant reduction across all conditionsfollowing the fatigue protocol. This decline was most pronounced within 50-75 %of the concentric phase relative to baseline measurements. Conclusion. Results indicate that the CMJ may be a useful tool for monitoring fatigue in at least 400m sprinters. These data also indicate that HIS may disproportionately reduce force output in duringconcentric movement. These insights may improve training prescriptions and injury prevention strategiesfor sprint athletes. 

PLEASE REVISE THE RESULTS SECTION ABOVE. There is Too much text. Drop narrative and just report data with p values. Only focus onmost salient data points.



The introduction is too long and doesn't flow. Authors should start with sprinting, then fatigue, then measures of fatigue (CMJ), then the purposes. The purposes need to be very clear and concise.While not uncommon to run studies without clear hypotheses, the perfect alignment with your findings and so much data suggests a search for data.
Results The results are toomuch. There is so much text its hard to follow. The authors need to consolidateto just what they tested first.

Most of these data can go into a table. Use figures to helpvisualize key points but don't present data in text, table AND figures. Useresults text to highlight key findings, table for lots of data, and fig for akey message. STREAMLINE!

Discussion 

The discussion is long and unfocused. The firstparagraph needs to restate the purposes then articulate which of the specifichypothese were targeted and hit. Right now it reads very post hoc. I’d like tosee the what you really expected and if you hit it. Then from those top 3 (?) aparagraph for each. One more paragraph or so to cover the other findings or speculation. 

Limitations. Tight paragraph to conclude. The take homemessage. Then finally applications.  The paper can really be trimmed down to bring themain message which is the CMJ detects fatigue and here’s why that matters. Too much text. Dropnarrative and just report data with p values. Only focus on most salient datapoints. The introduction is toolong and doesn't flow. Authors should start with sprinting, then fatigue, thenmeasures of fatigue (CMJ), then the purposes. The purposes need to be veryclear and concise. While not uncommon torun studies without clear hypotheses, the perfect alignment with your findings and so much data suggests a search for data.

Reviewer 3 ·

Basic reporting

The authors have taken into consideration the previous remarks and have significantly improved the clarity of the article. Indeed, the general introduction is well presented. It follows a good guiding thread that clues at the objectives and hypotheses. The important concepts are well defined, leaving little room for doubts and misunderstandings.

Experimental design

Introduction have highlighted limitations and gaps in the literature concerning the assessment of neuromuscular fatigue subsequent a sprint running effort.The reseach question is clear and well defined.

The method, results, and discussion are all divided into subsections that allow for a good understanding of the article.

Validity of the findings

The discussion has been greatly enriched, and the link between the results and their potential practical application has been well established.

---

## Round 0.3 · accepted · Accept

Dear authors:

After careful deliberation on your manuscript, we have considered accepting its publication in PeerJ. Congratulations on your work and thank you for trusting PeerJ.

Cordially,

Dr. Manuel Jimenez